# Using the NIH Research, Condition and Disease Categorization Database for research advocacy: Schizophrenia research at NIMH as an example

E. Fuller Torrey[1,2]*, Michael B. Knable[3,4], A. John Rush[5,6,7], Wendy W. Simmons[1], John Snook[8], D. J. Jaffe[9]†

1 Stanley Medical Research Institute, Kensington, Maryland, United States of America, 2 Uniformed Services University of the Health Sciences, Bethesda, Maryland, United States of America, 3 Clearview Communities, Frederick, Maryland, United States of America, 4 George Washington University Medical School, Washington, DC, United States of America, 5 Duke University School of Medicine, Durham, North Carolina, United States of America, 6 Texas Tech University, Lubbock, Texas, United States of America, 7 National University of Singapore, Singapore, Singapore, 8 Treatment Advocacy Center, Arlington, Virginia, United States of America, 9 MentalIllnessPolicyOrg, New York, New York, United States of America

† Deceased.
* torreyf@stanleyresearch.org

**Data Availability Statement:** All relevant data are within the manuscript and its Supporting information files.

## Abstract

In 2008 the National Institutes of Health established the Research, Condition and Disease Categorization Database (RCDC) that reports the amount spent by NIH institutes for each disease. Its goal is to allow the public "to know how the NIH spends their tax dollars," but it has been little used. The RCDC for 2018 was used to assess 428 schizophrenia-related research projects funded by the National Institute of Mental Health. Three senior psychiatrists independently rated each on its likelihood ("likely", "possible", "very unlikely") of improving the symptoms and/or quality of life for individuals with schizophrenia within 20 years. At least one reviewer rated 386 (90%), and all three reviewers rated 302 (71%), of the research projects as very unlikely to provide clinical improvement within 20 years. Reviewer agreement for the "very unlikely" category was good; for the "possible" category was intermediate; and for the "likely" category was poor. At least one reviewer rated 30 (7%) of the research projects as likely to provide clinical improvement within 20 years. The cost of the 30 projects was 5.5% of the total NIMH schizophrenia-related portfolio or 0.6% of the total NIMH budget. Study results confirm previous 2016 criticisms that the NIMH schizophrenia-related research portfolio disproportionately underfunds clinical research that might help people currently affected. Although the results are preliminary, since the RCDC database has not previously been used in this manner and because of the subjective nature of the assessment, the database would appear to be a useful tool for disease advocates who wish to ascertain how NIH spends its public funds.

**Funding:** The authors received no specific funding for this work.

**Competing interests:** I have read the journal's policy and the authors of this manuscript have the following competing interests: John Rush: A. John Rush has received consulting fees from Akili, Brain Resource Inc., Compass Inc., Curbstone Consultant LLC, Emmes Corp., Johnson and Johnson (Janssen), Liva-Nova, Mind Linc, Otsuka-US, Sunovion; speaking fees from Liva-Nova; and royalties from Guilford Press and the University of Texas Southwestern Medical Center, Dallas, TX (for the Inventory of Depressive Symptoms and its derivatives). He is also named co-inventor on two patents: U.S. Patent No. 7,795,033: Methods to Predict the Outcome of Treatment with Antidepressant Medication, Inventors: McMahon FJ, Laje G, Manji H, Rush AJ, Paddock S, Wilson AS; and U.S. Patent No. 7,906,283: Methods to Identify Patients at Risk of Developing Adverse Events During Treatment with Antidepressant Medication, Inventors: McMahon FJ, Laje G, Manji H, Rush AJ, Paddock S. This does not alter our adherence to PLOS ONE policies on sharing data and materials. The other authors have no disclosures or competing interests.

# Introduction

The National Institutes of Health is the largest source of funding for medical research in the world. As such, it is a major source of hope for individuals afflicted with specific diseases and their associated advocacy groups. In 1998, the Institute of Medicine (IOM) issued a report entitled "Scientific Opportunities and Public Needs" noting that there were problems regarding the public's perception of how research funds were allocated at NIH [1]. Specifically, it said that some believed "that NIH cares more about curiosity than cure, more about fundamental science than clinical application". The IOM report highlighted the need for public input into NIH using "a formal mechanism through which the public can inform the priority setting". One of the recommendations of the report was that "NIH should improve the quality and analysis of its data on funding by disease". Similar discussions subsequently took place among members of Congress, and in 2006, as part of the NIH Reauthorization Act, Congress mandated that "the Director of NIH shall establish an electronic system to uniformly code research grants and activities. . .The electronic system shall be searchable by a variety of codes, such as the type of research grant, the research entity managing the grant, and the public health area of interest" [2].

The result of this congressional mandate was the NIH Research, Condition, and Disease Categorization (RCDC) system, a publicly available online computerized database, updated annually, that reports the amount being spent by each NIH institute for each disease category. The RCDC includes extramural grants and contracts as well as intramural research projects for all 27 NIH institutes and centers. The database includes the name and institution of the principal investigator; the project's funding history; an abstract summarizing the research; and a statement regarding the project's public health relevance. The explicit intent of Congress was to make such information available to the public. As stated on the RCDC website: "The American people want to know how the NIH spends their tax dollars. The RCDC process categorizes the NIH research projects funded with those tax dollars" [3].

The RCDC database is thus a potentially rich research resource that could be useful for disease advocacy groups. Surprisingly, the database appears to be little known or used. A Medline search of articles related to the RCDC identified only two articles, on statistics [4] and disease burden [5]. Another article used the RCDC database without naming it to ascertain NIH research expenditures for cystic fibrosis and sickle cell disease [6]. The present report illustrates how the RCDC can be used by advocates to evaluate NIH research for particular diseases. We used schizophrenia research funded by the NIMH to illustrate how this information can inform the public about funding priorities.

Schizophrenia is one of the nation's most important diseases. Its one-year prevalence in the United States, as reported by the National Institute of Mental Health (NIMH) to Congress in 1993, is 1.5% among adults 18 and over and 1.2% among children 9 to 17 [7]. These numbers were based on the Epidemiologic Catchment Area study, the last in depth prevalence study carried out in the U.S. Based upon the estimated 2018 population of 327.2 million, this translates into 4.25 million Americans affected by this disease. Schizophrenia is also a major contributor to the problems of homelessness and the overcrowding of jails and prisons. The most recent estimate of the annual cost of schizophrenia in the United States is $155.7 billion [8].

In recent years, concerns have been raised regarding the relevance of schizophrenia research carried out by NIMH. An editorial published in the *British Journal of Psychiatry* in 2016, authored by 20 current or former members of the NIMH National Advisory Mental Health Council, including one of the present authors (Dr. Rush), noted that NIMH research had become increasingly focused on basic research, especially genetics and neural circuits, instead of more clinical research that might help people currently afflicted. They therefore

called "for an increase in public discussion of how to appropriate funding resources across mental health research domains" [9].

Others have expressed similar concerns. Dr. Steven Hyman, a former NIMH Director (1996–2001), recently claimed that "no new drug targets or therapeutic mechanisms of real significance have been developed for more than four decades" [10]. Dr. Thomas Insel, another former NIMH Director (2002 to 2015), observed that despite spending $20 billion "on the neuroscience and genetics of mental disorders. . . I don't think we moved the needle in reducing suicide, reducing hospitalization, [or] improving recovery for the tens of millions of people who have mental illness" [11].

Two advocacy groups in the United States focus specifically on serious mental illness, especially schizophrenia: the Treatment Advocacy Center in Arlington, Virginia (www.treatmentadvocacycenter.org) and Mental Illness Policy Org in New York City (www.mentalillnesspolicy.org). Other advocacy groups such as The National Alliance on Mental Illness (NAMI) and Mental Health America focus on mental disorders more broadly but do not specifically target schizophrenia. Based on the reports of NIMH's apparent failure to carry out sufficient clinical research (9–11), the authors undertook discussions with the two schizophrenia-focused advocacy groups regarding how to assess the NIMH research portfolio.

## Methods

The following research was not based on a formal protocol but rather evolved from the discussions cited above. One of the authors (reviewer one) became aware of the RCDC database and proceeded to do a rating of the projects to determine the feasibility of such research. Reviewer one then recruited reviewers two and three to independently do a similar rating, masked to the results of the other two assessors. All three reviewers had an association with one or both of the advocacy groups and shared a belief that NIMH should support a balanced portfolio between basic and clinical research, acknowledging that both were necessary. Each of the reviewers had had extensive clinical experience with patients with schizophrenia as well as extensive research experience, including having been funded by NIMH for research on schizophrenia and/or bipolar disorder. Dr. Rush served as both a member or Chair of three different research study sections at NIMH, as a member of the NIMH National Advisory Mental Health Council and as coauthor of the 2016 consensus statement referenced above [9].

Each reviewer was asked to estimate the relative likelihood (likely, possible, very unlikely) that the completion of the research project would improve the symptoms and/or quality of life for persons with schizophrenia during the next twenty years. The terms "likely", "possible", and "very unlikely" were not formally defined. Instead the reviewers relied on general medical knowledge regarding how long it has taken basic brain research in such areas as genetics and neural circuits to be translated into clinically useful knowledge. The twenty-year period was chosen based on the fact that the median age of all Americans is 38 and individuals with schizophrenia have a 15–25 year shortened life expectancy [12]. It was assumed that individuals affected with schizophrenia should have a reasonable chance of profiting from ongoing NIMH research during their remaining lifetime.

The public RCDC database was accessed for 2018, the most recent year for which final data was available at the time of the study. See accompanying box for details on accessing this system. In the RCDC system, disease relevance is ascertained by a computerized analysis of all funded proposals.

How to access the NIH schizophrenia–related research grants:

- Go to: https://report.nih.gov/categorical_spending.aspx.

- Scroll down the table until you find schizophrenia.

- Go to the 2018 column which has $248 million;

- Click on the $248 million. This will give you a listing of all 542 NIH schizophrenia–related grants for 2018.

- For details of any given grant click on the grant number. This will provide you with the project information, including an abstract, a statement on public health relevance, and the grant's funding history.

For 2018 the RCDC database listed 542 new and ongoing NIH-funded projects as being schizophrenia-related, which included 428 funded by NIMH and 114 funded by other NIH institutes. Of the latter, the largest number (n = 26) were basic brain studies funded by the National Institute of Neurological Disorders and Stroke (NINDS). Other examples of such grants and contracts included studies of smoking cessation in schizophrenia funded by the National Institute on Drug Abuse and the development of new Positron Emission Tomography (PET) ligands funded by the National Institute of Biomedical Imaging and Bioengineering.

The results of the independent ratings were summarized qualitatively and were also tested for inter-rater reliability using SAS software, version 9.4, to calculate the intraclass correlation coefficient for Fliess kappa. In reporting these results the Standard Reporting Qualitative Research (SRQR) guidelines were also used [13].

## Results

The NIMH schizophrenia-related research portfolio for 2018 consisted of 428 projects costing a total of $201 million, which represented 11.5% of NIMH's total budget of $1.755 billion for 2018. It included 418 extramural grants and contracts costing $176 million or an average of $421,000 per project per year. It also included $25 million for schizophrenia-related intramural projects which included eight separate laboratory projects, $3.9 million for the Office of the Intramural Scientific Director and $10.2 million for "NIMH space activation, maintenance and improvement". The information available in the abstracts on the RCDC database was sufficient for assessing most research projects but less so for the intramural projects or for research centers when multiple projects were summarized in a single abstract.

The average duration of funding for the 428 NIMH research projects in 2018 was four years (range 1–31 years). Thirty-eight extramural projects had been funded for 10 years or more and 11 of these for 20 years or more. The longest funded projects were two training grants funded respectively for 31 and 28 years along with a genetics study and a study of sensory processing, each funded for 27 years. Among the intramural projects a study of neuroimaging had been funded for 26 years.

Geographically, the 418 extramural awards went to 107 institutions. There was a significant clustering of awards with seven institutions receiving a total of 120 awards. The University of Pittsburgh received the most awards with 30, followed by Johns Hopkins University with 20. Columbia University, the University of Maryland, Icahn School of Medicine at Mount Sinai, and UCLA each received 15 or more. Only five awards were made to non-American principal investigators—four Canadians and one Swedish investigator—although a few others had foreign components administered by an American principal investigator.

Table 1 and Fig 1 summarize the assessments of the three individual reviewers regarding the likelihood of the research projects improving the symptoms and/or quality of life for persons with schizophrenia during the next 20 years. At least one reviewer rated 90% (386/428) of

**Table 1. Likelihood of research projects (n = 428) improving the symptoms and/or quality of life for persons with schizophrenia within 20 years.**

|  | Very unlikely | Possible | Likely |
|---|---|---|---|
| Reviewer 1 | 337 (79%) | 80 (19%) | 11 (2%) |
| Reviewer 2 | 341 (80%) | 67(16%) | 20(4%) |
| Reviewer 3 | 366 (86%) | 56(13%) | 6(1%) |

the research projects as being very unlikely to improve the symptoms and/or quality of life for persons with schizophrenia during the next 20 years, and all three reviewers agreed on this 71% (302/428) of the time.

Similarly, at least one reviewer rated 29% (126/428) of the research projects as possibly able to improve the symptoms and/or quality of life for persons with schizophrenia during the next 20 years. When the categories of "possible" and "likely" are combined, all three reviewers unanimously agreed that only ten percent (43/428) of the research projects held such promise. Overall, the intraclass correlation coefficient revealed a moderate level of agreement among the three raters (Fliess kappa = 0.473). Agreement was highest for those rated "very unlikely" (0.570); intermediate for those rated as "possible" (0.427); and poor for those rated as "likely" (0.165). This is also reflected in Fig 1.

Taken together, the individual reviewers selected a total of 30 projects (30/428) or 7% as being likely to improve the symptoms and/or quality of life for persons with schizophrenia within the next twenty years. Seven of these thirty were selected by two of the reviewers but none was selected by all three. For twenty-one of the thirty, all reviewers agreed that the project was either possible or likely. Examples of projects selected as likely included attempts to improve social function, medication adherence, symptoms such as auditory hallucinations, and clozapine usage; Table 2 lists all thirty research projects. The total cost of the thirty projects selected by one or more reviewer as likely to improve the symptoms and/or quality of life for persons with schizophrenia was $11,120,544; this was 5.5% of the schizophrenia-related research portfolio or 0.6%of the total NIMH budget for 2018. Note also that the average cost of a research project rated as likely was $371,000; this was significantly less than the $421,000 average cost for all extramural schizophrenia-related projects.

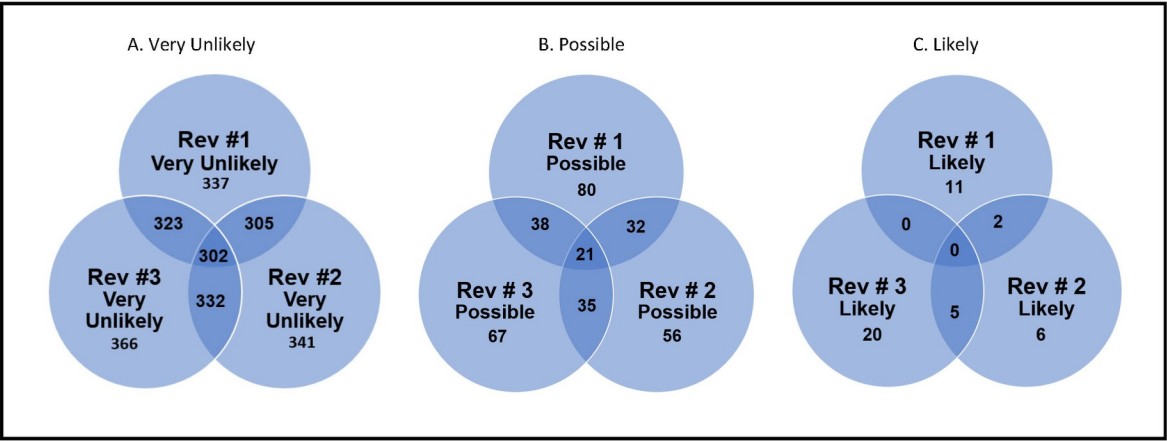

**Fig 1. Venn diagrams to indicate relationship amongst reviews' ratings of the research projects.**

**Table 2. The 30 research projects selected by at least one of the reviewers as likely to improve the symptoms and/or quality of life for persons with schizophrenia within 20 years.**

| | |
|---|---|
| CSC OnDemand: An Innovative Online Learning Platform for Implementing Coordinated Specialty Care | 4R44MH111283-02 |
| A transdiagnostic sleep and circadian treatment to improve community SMI outcomes | 5R01MH105513-04 |
| Testing Effectiveness of a Peer Led intervention to Enhance Community Integration | 5R01MH102230-04 |
| Imaging Neuroinflammation in Clinical high risk and Schizophrenia | 5R01MH100043-05 |
| Enhancing Social Functioning in Schizophrenia through Scalable Mobile Technology | 5R21MH111501-02 |
| Targeting Stress Reactivity in Schizophrenia: Integrated Coping Awareness Therapy | 5R33MH100250-05 |
| Dimensional outcomes and neural circuitry associated with psychosis risk | 1R01MH112584-01A1 |
| Creating Live Interactions to Mitigate Barriers (CLIMB): A Mobile Intervention to Improve Social Functioning in People With Schizophrenia | 1R43MH114765-01A1 |
| Longitudinal Mediation Analysis to Identify Effective Intervention Components in Clustered Trial of RAISE-ETP (Recovery after an Initial Schizophrenia Episode-Early Treatment Program) | 5R03MH112053-02 |
| Effectiveness of a Mobile Texting Intervention for People with Serious Mental Illness | 5R56MH109554-02 |
| Using mHealth to optimize pharmacotherapy regimens | 1P50MH115843-01 |
| Administrative Core for the following three research projects: | 1P50MH115842-01 |
| Adapting an evidenced-based weight management intervention and testing strategies to increase implementation in community mental health programs | 1P50MH115842-01 |
| Promoting evidenced-based tobacco smoking cessation treatment in community mental health clinics | 1P50MH115842-01 |
| Using an innovative quality improvement process to increase delivery of evidenced-based CVD risk factor care in community mental health organizations | 1P50MH115842-01 |
| Levetiracetamin in First Episode Psychosis | 5R61MH112833-02 |
| Texting for Relapse Prevention: Improving outcomes for people with schizophrenia | 5R34MH108781-03 |
| A Trial of "Opening Doors to Recovery" for Persons with Serious Mental Illnesses | 5R01MH101307-06 |
| Peer Support and Mobile Technology Targeting Cardiometabolic Risk Reduction in Young Adults with SMI | 5R01MH110965-03 |
| Real-time fMRI Neurofeedback as a Tool to Mitigate Auditory Hallucinations in Patients with Schizophrenia | 1R61MH113751-01A1 |
| Neural Biomarkers of Clozapine Response | 5K23MH110661-04 |
| Using Medicaid data to advance care for people with schizophrenia at risk for HIV (Medicaid-DASH) | 5R01MH112420-02 |
| Biomarker and Safety Study of Clozapine in Benign Ethnic Neutropenia | 5R01MH102215-04 |
| Trajectories of treatment response as window into the heterogeneity of psychosis: a longitudinal multimodal imaging study in medication-naieve first episode psychosis patients | 1R01MH113800-01A1 |
| Trial of Integrated Smoking Cessation, Exercise, and Weight Management in SMI | 5R01MH104553-05 |
| Targeting Auditory Hallucinations with Alternating Current Stimulation | 5R33MH105574-04 |
| ASSESSMENT OF MEDHERENT MEDICATION MANAGEMENT DEVICE AND ADHERENCE PLATFORM | 1R44MH116765-01 |
| Early Stage Identification and Engagement to Reduce Duration of Untreated Psychosis (EaSIE) | 5R34MH115463-02 |
| Molecular pathways of the kynurenine system in the neuroimmunology and psychophysiology of schizophrenia. | 1R21MH117512-01 |
| CRCNS: Collaboration toward an experimentally validated multiscale model of rTMS | 1R01MH118930-01 |

## Discussion

In keeping with the intent of Congress, the RCDC public database was used to assess the schizophrenia-related research projects funded by NIMH in 2018. Three experienced reviewers independently but unanimously agreed that 71% of the research projects were very unlikely

to improve the symptoms and/or quality of life for individuals with schizophrenia over the next 20 years. The reviewers also unanimously agreed that only 10% of the research projects held any possibility for such improvement. One or more reviewers selected 30 out of the 428 total research projects as actually likely to lead to such improvement but no project was so rated by all three reviewers. The results of the study would appear to confirm the opinion of 20 current or former members of the National Advisory Mental Health Council that the NIMH research portfolio is disproportionately focused on basic research instead of clinical research that might help people currently affected [9].

The information generated by this study will be useful to the schizophrenia-related advocacy groups that co-authored this study as well as to other advocacy groups and individuals with an interest in this research. The advocacy groups will encourage their followers to contact NIMH and their representatives in Congress. From past experience we have found that advocates for individuals with schizophrenia and other serious mental illnesses feel very strongly about the paucity of research on these diseases. When provided with specific information, they will often contact their congressional representatives as well as writing letters to newspapers. A follow up study using the RCDC database looking at NIMH's funding over time is already underway. The present study also demonstrates the feasibility of using the RCDC database to generate information of value to other disease advocacy groups. The authors plan to send the current study to the major disease advocacy groups in the United States to make them aware of this resource and its potential for influencing future research funding decisions by NIH. This is consistent with the intent of Congress in creating this database.

In assessing the NIMH schizophrenia-associated research portfolio, the reviewers were also impressed by the research areas being ignored. Foremost among these were attempts to find better treatments for individuals with schizophrenia. Among the 428 research projects there was just one treatment trial using a pharmacological agent, the anticonvulsant levetiracetamin, to treat individuals with schizophrenia. Another trial used amphetamine to enhance the effects of cognitive therapy and a third trial used a gluten-free diet. Two additional treatment trials used forms of transcranial stimulation in attempts to improve the symptoms of schizophrenia. These five trials together, representing 1.2% of the 428 research projects, were the only such trials.

We contend that NIMH should invest much greater resources to find better treatments for schizophrenia, including testing off label and potentially repurposed medications and other compounds for which there is no intellectual property protection, including prebiotics and probiotics. A major program should also be undertaken to find medications effective against schizophrenia's negative symptoms. Cost-benefit studies should be undertaken for clozapine, the most effective anti-psychotic but markedly underutilized in the U.S. Head to head efficacy studies for various long-acting injectable antipsychotic would also be very useful.

Associated with the paucity of treatment trials for schizophrenia in the NIMH research portfolio was the absence of clinically useful studies on the optimal use of antipsychotic drugs. Twenty years ago, NIMH funded the Clinical Antipsychotic Trials of Intervention Effectiveness (CATIE), which provided clinicians with useful information regarding the selection of antipsychotic medication. It was arguably the most important schizophrenia research funded by NIMH in the last two decades. Nothing similar has been subsequently funded by NIMH despite many important and outstanding questions regarding which antipsychotic or combination is likely to be effective for specific subgroups of patients. Pharmaceutical companies are reluctant to fund head-to-head antipsychotic studies so unless NIMH does so such studies will not be done. This situation is similar to the need for head-to-head drug trials for such disorders as hypertension and rheumatoid arthritis.

NIMH should also invest resources in improving the lives of individuals with schizophrenia who are homeless or incarcerated. For example, it should study the efficacy of assisted outpatient treatment (AOT) in reducing homelessness as suggested in preliminary studies. For individuals with schizophrenia who are incarcerated for major crimes, NIMH should study the relative effectiveness of conditional release, Forensic Assertive Community Treatment (FACT) teams, and Psychiatric Security Review Boards in reducing recidivism.

Another research area conspicuously absent from the 428 NIMH research studies was the epidemiology of schizophrenia, its treatment or treatment outcomes; there was not a single epidemiological study. This stands in marked contrast to Europe where epidemiological studies are generating useful hypotheses regarding schizophrenia's etiology, including marked differences in the incidence of schizophrenia in different parts of Europe [14] and in the incidence of schizophrenia among various immigrant groups [15,16]. The last major epidemiological study of schizophrenia funded by NIMH was the 1980s Epidemiologic Catchment Area (ECA) study; one finding from this study was a very low lifetime prevalence of schizophrenia among Mexican Americans in Los Angeles, a finding that was never followed up [17].

In summary, the main strengths of this project are the use of a novel NIH database and its demonstration as a disease advocacy tool; the use of experienced psychiatric reviewers; and the collaboration between the reviewers and the advocacy groups. The study's main limitation is the subjective nature of the review criteria.

## Recommendations

The 1998 Institute of Medicine report on NIH research suggested the need for "a formal mechanism through which the public can inform the priority setting" [1]. The RCDC database provides this mechanism. Accordingly, the authors suggest the following.

1. Publicize the availability and encourage the use of the RCDC database for research advocacy groups for all medical and psychiatric disorders.

2. At least 50% of NIMH schizophrenia-related projects should be classifiable by advocacy groups as possible or likely to improve the symptoms and/or quality of life for individuals with this disease in less than 20 years.

3. Put a much greater funding priority on developing better neurobiological or psychological treatments for schizophrenia and/or enhance their delivery in clinical care settings. Include especially the many off-label medications and plant-derived compounds that may be useful in treating schizophrenia and for which there is insufficient intellectual property protection for these compounds to be tested by industry.

4. Request that the National Academy of Medicine convene a review group to examine the NIMH intramural research program and assess both its mission and its cost effectiveness. In 2018, according to the RCDC database, it included eight schizophrenia-related research projects at a cost of $25 million.

5. Incentivize public and private community healthcare systems to conduct research to address the clinical needs of patients and providers in order to bring basic/translational research results into practice more rapidly.

In conclusion, the NIMH schizophrenia research portfolio is disproportionately weighted toward basic research that promises better treatments in the distant future. One is reminded of the advice given in similar circumstances in 1971 by a leading cancer researcher: "We cannot

afford to sit and wait for the promise of tomorrow so long as stepwise progress can be made with the tools at hand today" [18].

## Supporting information

**S1 File. NIMH schizophrenia related research projects for 2018: 1 = likely; 2 = possible; 3 = unlikely.**
(XLSX)

**S2 File. Reporting checklist for qualitative study.**
(DOCX)

## Acknowledgments

We thank Dr. Robert H. Yolken for his help in planning this project and Corrie Brown for her help with the Venn Diagrams. We also thank Dr. Thomas Carmody for his assistance with the statistics.

## Author Contributions

**Conceptualization:** E. Fuller Torrey, John Snook, D. J. Jaffe.

**Data curation:** Wendy W. Simmons.

**Formal analysis:** Michael B. Knable.

**Investigation:** E. Fuller Torrey, Michael B. Knable, A. John Rush.

**Methodology:** E. Fuller Torrey, Michael B. Knable, A. John Rush.

**Project administration:** E. Fuller Torrey, Wendy W. Simmons.

**Resources:** Wendy W. Simmons.

**Writing – original draft:** E. Fuller Torrey.

**Writing – review & editing:** Michael B. Knable, A. John Rush, Wendy W. Simmons, John Snook, D. J. Jaffe.

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
