## [Decision Letter · Decision Letter 0]

15 Jul 2020

PONE-D-20-13102

Using the NIH Research, Condition and Disease Categorization Database for research advocacy: Schizophrenia research at NIMH as an example

PLOS ONE

Dear Dr. Torrey,

Thank you for submitting your manuscript to PLOS ONE. After careful consideration, we feel that it has merit but does not fully meet PLOS ONE’s publication criteria as it currently stands. Therefore, we invite you to submit a revised version of the manuscript that addresses the points raised during the review process.

First, I would like to thank the two authors for their important comment. Both suggest that the papers has merits and I agree. 

I have however some additional comments that need to be addressed : 

- First : the manuscript must mention explicitly : 

. If there was a protocol YES / NO ; 

. If it was registered a priori (e.g. on the Open Science Framework) ; 

. Data must be submitted as a supplementary material ; 

. The paper should be reported using the adequate reporting guideline (and a specific grid should be filled and provided): the method section must be expanded to allow for reproduction of the study ; 

. A limitation of the study is the subjectivity in the assessment and the fact that the outcome used in this study had no prior validation : the exploratory nature must be explicit in both the limitation section and the abstract to avoid any spin and to avoid any over-interpretation of the results ; 

. I suggest drawing a figure using Venn Diagram (or an alluvial plot) to show the agreement among the 3 reviewers. I acknowledge that it can be challenging to draw such a figure, but this is all the more important ; 

. I also suggest that the full list of proposals listed as likely is dysplayed in a table in the main manuscript ; 

. The same in a web appendix for studies listed as possible and for those listed as unlikely ; 

We look forward to receiving your revised manuscript.

Kind regards,

Florian Naudet, M.D., M.P.H., Ph.D.

Academic Editor

PLOS ONE

Journal Requirements:

2. We note that you have a patent relating to material pertinent to this article.

a. Please provide an amended statement of Competing Interests to declare this patent (with details including name and number), along with any other relevant declarations relating to employment, consultancy, patents, products in development or modified products etc.

Please confirm that this does not alter your adherence to all PLOS ONE policies on sharing data and materials, as detailed online in our guide for authors http://journals.plos.org/plosone/s/competing-interests by including the following statement: "This does not alter our adherence to  PLOS ONE policies on sharing data and materials.” If there are restrictions on sharing of data and/or materials, please state these.

Please note that we cannot proceed with consideration of your article until this information has been declared.

Reviewers' comments:

Reviewer's Responses to Questions

**Comments to the Author**

1. Is the manuscript technically sound, and do the data support the conclusions?

Reviewer #1: Partly

Reviewer #2: Yes

2. Has the statistical analysis been performed appropriately and rigorously? 

Reviewer #1: I Don't Know

Reviewer #2: Yes

3. Have the authors made all data underlying the findings in their manuscript fully available?

Reviewer #1: Yes

Reviewer #2: Yes

4. Is the manuscript presented in an intelligible fashion and written in standard English?

Reviewer #1: Yes

Reviewer #2: Yes

5. Review Comments to the Author

Reviewer #1: This is an interesting paper on an important topic for advocacy organisations and consumer participants. I would like to have seen more emphasis on the public perspective in the paper and how advocacy groups had informed the design of the study, how they could influence the development and use of the RCDC given the conclusions of the authors. It was not clear whether this review was attempting to demonstrate that the database was not functional, or the research recorded in it was not of sufficient value. The paper appears to be arguing two perspectives, firstly that the RCDC could be a useful tool and secondly that the schizophrenia research within it is not providing value for money and is not of sufficient efficacy to lead to improvements in patient outcomes. At present the focus of the paper seems to be a little more on the authors area of interest i.e. the relevance of the schizophrenia research funded by the NIH, than the stated focus of the study which is the use of the database for research advocacy. It would be useful if the authors could state their intentions more clearly. They need to be clearer about the messaging and intent of the study. The key message, which is implied at the beginning and the end of the paper, seems to be that the public needs to inform research priorities and that the RCDC could be a valuable tool in facilitating this.

The intended purpose of creating the RCDC database was not entirely clear from the paper. It appears to be simply to provide a source of information for the public. Was it also the intention to engage the public and/or advocacy groups in the prioritisation of research topics? If the primary intention of the RCDC database was to enable public access, and presumably increase the degree of transparency of NIH funding decisions, it would be interesting to know if the authors considered including advocacy organisations as reviewers in the study or if there is any intention for them to carry out a similar review following this study to evaluate the value of the database in that context. The paper refers to how the RCDC might be a rich research resource for advocacy groups but does not explain how those groups might use it. For example, would or can they use it to identify unanswered research questions (in a similar way to the James Lind Alliance), to identify ongoing or completed research, or to identify potential partners for research? How does this then inform the public about funded research?

The outline of the methodology would benefit from greater detail. The database was accessed for data for 2018; however, it is not clear if the study was reviewing projects approved and funded in 2018 or all the projects that were ongoing in 2018 but may have started prior to that date. This is important information in the context of the discussion in the results section relating to the clustering of awards and whether there may have been an historical shift in the location of awards or changes in the types of projects funded. It would also be helpful to understand what criteria the three assessors used for their ratings of ‘likely’, ‘possible’ or ‘very unlikely’. For example:

- Was it based on their individual experience and expertise, knowledge of clinical decision making or did they develop a shared assessment framework?

- How was the decision to have three assessors arrived at? Did this give the study sufficient power?

- How did the reviewers ensure any potential bias towards the topic areas was excluded?

- The reviews seem to be based on an assessment of project abstracts, were they sufficiently detailed to make an assessment of impact over a period of 20years?

- Why was the period of 20 years selected as he optimum period for the demonstration of clinical improvement?

Specific comments relating to the content of the article:

- The term ‘allegations’ in the abstract seems inappropriate and unnecessarily adversarial. It could be expressed as ‘criticisms’. The authors also need to substantiate their statement to identify which stakeholder(s) made the ‘allegations’.

- The final sentence of the abstract needs to be qualified as it is a very broad statement which reads as the personal opinion of the authors rather than something substantiated by their study. Did the study show that ‘the RCDC database is an excellent tool’ and should be used more frequently, or is this the authors view?

- The statement on page 4 that refers to the database being little known or used, cites academic articles that reference the RCDC. Presumably the articles do not report data about the number of public individuals who search the database or use the information from it as a lobbying tool for their disease area? What is the level of public access to the database? Advocacy groups, who are users of research, are less likely to publish academic articles or lead research therefore their use of the database may not be referenced in past papers.

- Page 4 refers to the prevalence of schizophrenia in the US; how does this compare with the prevalence of other disease areas and the cost to the economy?

- Some abbreviations and acronyms are not explained e.g. NAMI, PET.

Without a more detailed description and rational for the review process, the paper is weakened. It is important to have this detail if the study is to be replicated in other disease areas as suggested by the authors. Occasionally the paper seems to reflect the personal perspective of the authors in relation to the types of research that receive funding. To provide a more rounded perspective, it would be useful to balance this with the types of research clinicians report as important and the research topics of importance to advocacy groups and patients. It is not clear how, or if, patients inform priority setting although this was a stated intention of the I of M report on NIH research. I would like to see more emphasis on the public perspective, and how advocacy groups can inform the development and use of the RCDC, within the paper.

Reviewer #2: This article reports on the percentage and associated expenditures of the research portfolio funded by the USA’s National Institute of Mental Health in the area of schizophrenia and provides an assessment of the extent of the portfolio that is likely to provide clinical improvement to individuals with the disorder over the next 20 years. The article is very well written, balanced, and timely. It has numerous strengths, including a succinct but informative review of the push by advocacy organizations for public accountability in Congressionally funded mental health research, the public-health significance of schizophrenia, and the NIMH research portfolio for 2018. Other strengths include the use of a multi-rater system for coding the likelihood of clinical impact, estimation of an intra-class correlation coefficient among raters, discussion of the areas of research being ignored, and a cogent set of recommendations. The ms. also has a few minor weaknesses, which would strengthen the article if addressed:

• P.4, lines 77-78: Some readers may not understand why the ms. goes back to 1993 for prevalence data on schizophrenia in the USA. Aren’t more recent data available, they may ask? One advantage of these data is that they were provided by the NIMH itself to Congress, thereby making them the “official” rates of the organization. In addition, as the ms. points out, more recent community-based epidemiological data on schizophrenia in the USA are limited, due in part to methodological difficulties and lack of funding interest. The ms. might benefit from a sentence that clarifies why 27-year-old data are being used to report the national prevalence of the disorder.

• P.8, lines 152-154: The ms. usefully notes the methodological limitation of more sparse descriptive information on research studies conducted in the intramural program and research centers. However, it does not state how this limitation was addressed, if at all. One possibility is to provide percentages of the NIMH budget spent on schizophrenia research separately by type of research program (i.e., extramural, intramural, research center).

• The ms. might be strengthened with a few more examples of the kind of schizophrenia research the NIMH could fund (and proactively request) that is likely to be of more immediate benefit. Some examples are mentioned (e.g., new medications and psychological treatments) but more information and additional examples would make the article more compelling by noting the potential benefits that are not being reaped. Otherwise, some readers may think the NIMH is following its policy because there is little to study that could be more immediately useful.

6. PLOS authors have the option to publish the peer review history of their article (what does this mean?). If published, this will include your full peer review and any attached files.

Reviewer #1: **Yes: **Dr Virginia Minogue

Reviewer #2: No

---

## [Author Response · Author response to Decision Letter 0]

3 Aug 2020

Dear Dr. Naudet, Dr. Minogue and Reviewer #2, 

Thank you for your useful comments and critiques. We have done the following: 

• We have stated explicitly that there was not a protocol.

• The master file, including all 428 research projects, by title and number along with the grades of the three reviewers are being submitted as supplementary material. To make the order of reviewers consistent between supplementary material and the manuscript, we reversed the numbers between reviewers 2 and 3 in Table 1.

• We have included a reporting checklist, along with mention in the methods section, a reference, and a specific grid.

• We have re-written the methods section to make explicit the exploratory nature of the study.

• We have also included this in the sections on limitations at the end of the paper. 

• We have included three Venn diagrams to show the agreement among the three reviewers for the three questions. 

• We have added Table 2 to include the full list of the 30 proposals selected as “likely”.

• The supplementary material now includes all studies in the same order as they are listed in the RCDC database. Thus individuals can ascertain the rating for all studies listed as “likely”, “possible”, or “very unlikely”. 

• The Venn Diagrams are being submitted as separate figure files. 

• The Title and Numbers of the two patents that were included pertain to depression and thus are not relevant to our study on schizophrenia. We have added the suggested statement.

Dr. Minogue

• In re-writing the methods section, we clarified the intent and development of the study, including the use of the advocacy groups in its development.

• The intended purpose of the RCDC database was to make the information available to the public and to shift the lobbying for funding from members of Congress to the NIH institutes. The directors of the advocacy groups were included as authors.

• We have added a section in the discussion regarding how the study will be used by the advocacy groups. Unfortunately we do not have an organization similar to the James Lind Alliance as far as I know. The main relationship between the NIH institutes and most disease advocacy groups is the former asking the latter to ask Congress to give them more money; thus what we are doing is very different.

• The methods section has been enlarged to provide more details on the development of the study. We have added “new and ongoing” to indicate that the projects from the past have been included. 

• We have clarified that the ratings of “likely”, “possible”, and “very unlikely” were subjective based on individual experience and expertise. We did not attempt to standardize the definitions. 

• We decided to use three assessors as a guess, based on the fact that we had 428 research projects. We had no idea when we started how much agreement we would have and were pleasantly surprised to find that we had good agreement.

• We added a sentence to clarify that our shared bias was that NIMH should be doing both basic and clinical research. 

• Most of the project abstracts were sufficiently detailed to make a reasonable assessment of what they are likely to produce over the next twenty years. 

• We have added an explanation for why twenty years was selected as the optimum period.

• In the abstract “criticisms” has been substituted for “allegations”.

• The final sentence in the abstract has been modified.

• Based on my discussion with the NIH official who was in charge of the RCDC until he recently retired, I think it is very unlikely that the database was used by any disease advocacy groups.

• The prevalence of schizophrenia in the US is probably a little above average by world standards. In most studies, it has been said to be one of the costliest diseases.

• We have spelled out NAMI and PET.

• We have significantly enlarged the description for the review process.

• We have enlarged the discussion of research projects that would be clinically useful that NIMH should be doing. 

Reviewer #2

• We have clarified why we are using the 1983 prevalence data. 

• Although the descriptions of some of the intramural research projects was sparse, the reviewers had independent information on many of them. For example, two of us had worked with researchers doing the imaging studies many years ago. Thus we felt comfortable in our ratings. The percentage of NIMH funds going to intramural projects is stated at the beginning of the results section. 

• We added additional examples of the kind of schizophrenia research that NIMH should be funding.

---

## [Decision Letter · Decision Letter 1]

27 Aug 2020

PONE-D-20-13102R1

Using the NIH Research, Condition and Disease Categorization Database for research advocacy: Schizophrenia research at NIMH as an example

PLOS ONE

Dear Dr. Torrey,

Thank you for submitting your manuscript to PLOS ONE. After careful consideration, we feel that it has merit but does not fully meet PLOS ONE’s publication criteria as it currently stands. Therefore, we invite you to submit a revised version of the manuscript that addresses the points raised during the review process.

First, I would like to thank here the two reviewers who helped me in reaching a decision. 

I have some additional comments : 

- In the abstract, please make it more clear that the conclusions are only exploratory ; 

- Please add in the abstract a few words about the limitations, to avoid any spin / and a few words about the agreement ;

- By the may the agreement was not good for projects labeled as "Likely usefull" (see the Venn Diagram). Please avoid any spin about this and state it explicitely (it is not satisfying to see that you wrote that the agreement was good, especially for the unlikely category / one could say the agreement was not so good, especially for the likely category). Please edit both the text and the abstract.

- I tend to think that a Kappa would be more easy to interpret and indeed more appropriated to judge about inter-rater reliability than a Cronbach's alpha. Please change / or in case you disagree / please give me more reasons to think that you are right. 

We look forward to receiving your revised manuscript.

Kind regards,

Florian Naudet, M.D., M.P.H., Ph.D.

Academic Editor

PLOS ONE

Reviewers' comments:

Reviewer's Responses to Questions

**Comments to the Author**

1. If the authors have adequately addressed your comments raised in a previous round of review and you feel that this manuscript is now acceptable for publication, you may indicate that here to bypass the “Comments to the Author” section, enter your conflict of interest statement in the “Confidential to Editor” section, and submit your "Accept" recommendation.

Reviewer #1: All comments have been addressed

Reviewer #2: All comments have been addressed

2. Is the manuscript technically sound, and do the data support the conclusions?

Reviewer #1: Yes

Reviewer #2: Yes

3. Has the statistical analysis been performed appropriately and rigorously? 

Reviewer #1: Yes

Reviewer #2: Yes

4. Have the authors made all data underlying the findings in their manuscript fully available?

Reviewer #1: Yes

Reviewer #2: Yes

5. Is the manuscript presented in an intelligible fashion and written in standard English?

Reviewer #1: Yes

Reviewer #2: Yes

6. Review Comments to the Author

Reviewer #1: (No Response)

Reviewer #2: (No Response)

7. PLOS authors have the option to publish the peer review history of their article (what does this mean?). If published, this will include your full peer review and any attached files.

Reviewer #1: **Yes: **Dr Virginia Minogue

Reviewer #2: **Yes: **Roberto Lewis-Fernandez, MD

---

## [Author Response · Author response to Decision Letter 1]

10 Sep 2020

Florian Naudet, M.D., M.P.H., Ph.D.

Academic Editor

PLOS ONE

September 10, 2020

Dear Dr. Naudet, 

Thank you for the further review of our manuscript. We have done the following.

• Clarified in the abstract that the conclusions are only exploratory;

• Clarified in the abstract the limitations of the research, including the limitations of the agreement among the raters;

• Stated explicitly in the text that the interrater agreement was good for the “very unlikely” research projects, but only fair for the “possible” projects and poor for the “likely” projects. This should also be clear to readers from looking at the Venn diagrams.

Regarding the proper statistical test for assessing the interrater reliability, my colleagues (who understand this issue much better than I do) believe that the Cronbach's alpha is the more appropriate test because it is a measure of internal consistency and scale reliability. What we are testing is scale reliability, not a diagnostic test or x-ray, etc. Cronbach’s is most commonly used for scale reliability and that seems consistent with what we are doing. Cohen`s kappa, by contrast, is usually used when you have just two raters. However if you prefer that we use a kappa statistic we can look for an alternative to the Cohen’s kappa.

I am also sad to report that one of our authors, DJ Jaffe, died unexpectedly on August 23. I have so indicated on the title page.

E. Fuller Torrey, M.D

Associate Director for Research

Stanley Medical Research Institute

Kensington, MD

---

## [Editor Report · Decision Letter 2]

11 Sep 2020

PONE-D-20-13102R2

Using the NIH Research, Condition and Disease Categorization Database for research advocacy: Schizophrenia research at NIMH as an example

PLOS ONE

Dear Dr. Torrey,

Thank you for submitting your manuscript to PLOS ONE. After careful consideration, we feel that it has merit but does not fully meet PLOS ONE’s publication criteria as it currently stands. Therefore, we invite you to submit a revised version of the manuscript that addresses the points raised during the review process.

I'm really sad to learn that one of your co-authors died unexpectedly. I send you my sincere condolences. 

Thank you for the changes you made, the better is better now. But I'm still not convinced about the use of Cronbach' alpha in this occasion. You are not working on a multiple item scale. There are actually alternative to Cohen's Kappa such as Fleiss' kapa for instance, that can handle multiple rater. Please ask the help of a statistician to choose the approach that fits better your needs. 

We look forward to receiving your revised manuscript.

Kind regards,

Florian Naudet, M.D., M.P.H., Ph.D.

Academic Editor

PLOS ONE

---

## [Author Response · Author response to Decision Letter 2]

6 Oct 2020

RE: Manuscript PONE-D-20-13102R2

Using the NIH Research, Condition and Disease Categorization Database for research advocacy: Schizophrenia research at NIMH as an example

Florian Naudet, M.D., M.P.H., Ph.D.

Academic Editor. DATE: October 5, 2020

PLOS ONE

Dear Dr. Naudet, 

Following up your letter of September 11, we have consulted with Thomas Carmody PhD, a Professor in the Department of Population and Data Sciences at the University of Texas Southwestern. After reviewing our manuscript and some of the literature on interrater reliability, he concluded that both Fliess kappa and Cronbach alpha have been used in situations analogous to ours. He sent a couple of references justifying the use of the latter. However, he also said that the kappa “is more widely used for interrater agreement”, whereas alpha “is most often used to measure internal consistency of scales”. We, therefore, decided to use the Fliess kappa which he calculated for us:

 Fleiss kappa for each level of rating and overall

Rating Kappa Error z Prob > z

1 0.16513 0.027907 5.9171 <.0001

2 0.42658 0.027907 15.2858 <.0001

3 0.56954 0.027907 20.4083 <.0001

overall 0.47261 0.024540 19.2588 <.0001

We have made the appropriate changes to the text. 

Thank you for your assistance.

 Please let us know if you need additional information. 

E. Fuller Torrey MD

---

## [Editor Report · Decision Letter 3]

8 Oct 2020

Using the NIH Research, Condition and Disease Categorization Database for research advocacy: Schizophrenia research at NIMH as an example

PONE-D-20-13102R3

Dear Dr. Torrey,

We’re pleased to inform you that your manuscript has been judged scientifically suitable for publication and will be formally accepted for publication once it meets all outstanding technical requirements.

Thank you for all the edits that have improved the quality of the paper in my opinion. 

Kind regards,

Florian Naudet, M.D., M.P.H., Ph.D.

Academic Editor

PLOS ONE
---

## [Editor Report · Acceptance letter]

13 Oct 2020

PONE-D-20-13102R3 

Using the NIH Research, Condition and Disease Categorization Database for research advocacy: Schizophrenia research at NIMH as an example 

Dear Dr. Torrey:

I'm pleased to inform you that your manuscript has been deemed suitable for publication in PLOS ONE. Congratulations! Your manuscript is now with our production department. 

Kind regards, 

on behalf of

Pr. Florian Naudet 

Academic Editor

PLOS ONE